# High Intensity Interval Training Does Not Have Compensatory Effects on Physical Activity Levels in Older Adults

**DOI:** 10.3390/ijerph17031083

**Published:** 2020-02-08

**Authors:** Paolo Bruseghini, Enrico Tam, Elisa Calabria, Chiara Milanese, Carlo Capelli, Christel Galvani

**Affiliations:** 1Department of Molecular and Translational Medicine, University of Brescia, 25123 Brescia, Italy; 2Exercise and Sport Science Degree Course, Università Cattolica del Sacro Cuore, 20162 Milan, Italy; 3Department of Neuroscience, Biomedicine and Movement, School of Exercise and Sport Sciences, University of Verona, 37131 Verona, Italy; enrico.tam@univr.it (E.T.); elisa.calabria@univr.it (E.C.); chiara.milanese@univr.it (C.M.); carlo.capelli@univr.it (C.C.); 4Applied Exercise Physiology Laboratory, Department of Psychology, Università Cattolica del Sacro Cuore, 20162 Milan, Italy; christel.galvani@unicatt.it

**Keywords:** ageing, physical activity, high-intensity interval training, energy expenditure, lifestyle, elderly

## Abstract

Background: Exercise has beneficial effects on older adults, but controversy surrounds the purported “compensatory effects” that training may have on total daily physical activity and energy expenditure in the elderly. We wanted to determine whether 8 weeks of high-intensity interval training (HIIT) induced such effects on physical activity and energy expenditure in healthy, active older adult men. Methods: Twenty-four healthy elderly male volunteers were randomized to two groups. The experimental group performed HIIT (7 × 2 min cycling repetitions, 3 d/w); the control group performed continuous moderate-intensity training (20–30 min cycling, 3 d/w). Physical activity and energy expenditure were measured with a multisensor activity monitor SenseWear Armband Mini. Results: During HIIT, significant changes were observed in moderate and vigorous physical activity, average daily metabolic equivalents (METs), physical activity level, and activity energy expenditure (*p* < 0.05) but not in total energy expenditure. Sleep and sedentary time, and levels of light physical activity remained constant during the training period. Conclusions: The findings suggest that HIIT induced no compensatory effect: HIIT does not adversely affect lifestyle, as it does not reduce daily energy expenditure and/or increase sedentary time.

## 1. Introduction

Regular exercise has beneficial effects on health in older adults: exercise training can improve muscle function, increase aerobic capacity, and reduce the incidence of cardiovascular events and non-cardiovascular chronic disease, thus reducing physical frailty and delaying physical dependence [1,2]. Indeed, any increase in physical activity, no matter how small, can improve health [3]. The benefits of increasing regular physical activity have been observed in people irrespective of age, sex, ethnicity, or weight status [4]. Aging is associated with a decline in physical activity, and therefore exercise programs for the elderly are promoted to increase levels of physical activity [5].

High-intensity interval training (HIIT) consists of bouts of high-intensity exercise interspersed with active recovery phases. High-intensity exercise is not commonly prescribed in clinical practice because it may not be tolerated by non-trained subjects and because it may potentially expose them to increased risk of injury [6]. However, Wisløff and colleagues reported that the key factor in determining cardiovascular benefits is exercise intensity rather than duration [7,8]. As older adults report lack of time for physical activity, high-intensity interval training has been highlighted as a promising aerobic intervention in older adults due to its short duration. Previous studies investigating its effects on exercise capacity and metabolic risk factors have indicated that it can increase maximal oxygen uptake ($\overset{\cdot }{V}$O_2max_) more than traditional, continuous moderate-intensity exercise, with the same caloric expenditure, in trained men, sedentary subjects, and older people [9,10,11,12,13]. HIIT elicited similar or higher enjoyment and adherence levels than moderate-intensity continuous training [14]. Furthermore, it has been observed that high-intensity interval training improved body composition (fat and lean masses), muscle function (strength and power), or reduced cardiometabolic risk factors (waist circumference, waist–hip circumference ratio, blood pressure, and fasting glucose levels) in older adults [15,16,17,18,19].

The increase in physical activity through training may bring about lifestyle changes for more active subjects. Studies examining the extent to which exercise interventions can modify total energy expenditure (TEE) have reported that some people reduce their level of physical activity when they participate in an exercise workout (aerobic or strength training) [20,21]. This change in daily energy expenditure results in a saving of metabolic energy by altering appetite or adopting behavioral changes to reduce levels of physical activity and increase sedentary time [22,23,24,25,26,27]. Conversely, some studies found no compensatory effect in older adults after structured moderate or vigorous aerobic exercise or after resistance training, respectively [28,29,30]. 

Controversy surrounds the effects that training may have on total daily physical activity or TEE in older adults. To our best knowledge, no studies to date have analyzed the compensatory behavior of older adults during a HIIT-based exercise program. With this study, we wanted to determine whether 8 weeks of HIIT affected the level of physical activity and TEE in healthy older men.

## 2. Materials and Methods

### 2.1. Subjects

For this study 24 healthy, elderly male volunteers were recruited from among local residents in the Verona metropolitan area (Italy). Table 1 presents the characteristics of the study sample. Inclusion criteria were age 65–75 years, no osteo-articular diseases, normal electrocardiogram (ECG) at rest, no cardiovascular, metabolic, or respiratory diseases. A preliminary medical examination with a cycle-ergometer stress test was conducted to exclude abnormal response to exercise. The investigation was conducted in accordance with the ethical standards, with the Declaration of Helsinki, according to national and international guidelines and was approved by the authors’ institutional review board (approval on June 18th 2013, prot. no: 30191/09.11.01). Written, informed consent was obtained from all subjects.

### 2.2. Study Design

Subjects were randomly assigned to one of two groups according to an online randomization system (random.org). The experimental group (*n* = 12, EXP) performed HIIT; the control group (*n* = 12, CTRL) performed continuous moderate-intensity training. Aerobic performance and anthropometrical variables were evaluated before the beginning of training (PRE) and at the end of the 8-week training period (POST). During a brief test session before the baseline evaluation, all subjects were familiarized with the study procedures. The start of the training program was split so that, over a period of six subsequent days, subjects could, at random, begin training. Subjects were asked to maintain their usual eating and physical activity habits for the duration of the study. No dropouts were recorded during the study period. 

Physical activity was measured by means of a SenseWear Armband Mini (BodyMedia, Inc., Pittsburgh, PA, USA) worn for 1 week during the 2 months before the start of training (T1), 1 week during training (T2) (randomly during weeks 5, 6, and 7 of training), and then for 1 week during 2 months after the end of training (T3) (Figure 1) [31,32,33].

### 2.3. Anthropometry and Body Composition Assessment

Height, waist circumference, and hip circumference were measured from 3D human scans (Breuckmann BodySCAN^®^, Breuckmann GmbH, Meersburg, Germany). Whole body fat (WB fat, %) was measured by means of dual-energy X-ray absorptiometry (QDR Explorer W, Hologic, Marlborough, MA, USA). All procedures were executed in controlled conditions by the same examiner and as previously described in Bruseghini et al. [10]. Body-mass index (BMI, weight in kg divided by height in m^2^), waist-to-hip ratio (WHR, waist measurement divided by hip measurement), and waist-to-height ratio (WHtR, waist circumference divided by height) were recorded.

### 2.4. Cardiorespiratory Fitness Assessment

The procedure has been described previously [10], and only the main procedures are given below. $\overset{\cdot }{V}$O_2max_ was evaluated on an electro-magnetically braked cycle ergometer (Excalibur Sport, Lode, Groningen, The Netherlands). The ramp incremental test consisted of 3 min at rest and 3 min at 30 W followed by a continuous increment of the workload by 10–15 watts/min, depending on the prospective training status of each subject, until voluntary exhaustion. The incremental test was followed by a verification phase to exhaustion [34]. Expired gases were continuously collected for the determination of gas exchange on a breath-by-breath basis (Quark b^2^, Cosmed, Rome, Italy). Each test was preceded by a warm-up of 12 min (4 min at 30 W, 4 min at 45 W, 4 min at 60 W). The exercise protocol was performed at a room temperature of 18–20 °C. Subjects were instructed to refrain from strenuous physical activity for the 24 h before and from intake of caffeinated beverages on the day of the test session.

### 2.5. Physical Activity and Energy Expenditure Assessment

Daily physical activity and energy expenditure were measured with a SenseWear Armband Mini (SWAM). Subjects were asked to wear the SWAM 24 h a day, except during water-based activities, for six consecutive days including weekends: four weekdays and two days on the weekend [31,32]. On the weekdays during T2, the SWAM was worn on two training days and on two non-training days. The SWAM is a multisensor body monitor that is worn over the triceps muscle of the left arm. It enables continuous collection of various physiological and movement parameters through multiple sensors, including a tri-axis accelerometer and sensors measuring heat flux, galvanic skin response, skin temperature, and near body ambient temperature. Data from the sensors are combined with age, body weight, and height to estimate physical activity and energy expenditure levels. The data were processed by the latest software version v8.0 (algorithm v5.2) developed by the manufacturer (InnerView Professional Research Software, BodyMedia, Pitsburgh, PA, USA). The SenseWear Armband has been validated for estimation of energy expenditure and total sleep time [31,35]. Although it has been reported to underestimate energy expenditure when compared to doubly labelled water, it has been used as an appropriate criterion measure to evaluate the time spent in physical activity in older adults [33,36,37]. Data were calculated and downloaded in 1-min epochs. SWAM data were acceptable for analysis if the overall wear time was ≥85% of the total time the subjects had the SWAM in their possession [31].

In addition, the subjects were asked to record their activities in a diary: this information served to substitute missing data when the SWAM was not worn. Missing data for activities of personal care and swimming were calculated based on a constant metabolic equivalents MET value according to the 2011 Compendium of Physical Activities [38]. Missing values for sleep were imputed from the mean MET value of sleep recorded on all other nights.

Physical activity was divided into four intensity categories: sedentary (SED, min/d, <1.5 METs), light intensity (LPA, min/d, 1.5–2.9 METs), moderate intensity (MPA, min/d, 3–6 METs), and vigorous intensity (VPA, min/d, >6 METs) [39]. Sleep duration (Sleep, min/d) was calculated using an algorithm based on the heat-flux sensor and the accelerometer [40]. The sum of moderate and vigorous physical activity was defined as moderate-to-vigorous intensity physical activity (MVPA, min/d, >3 METs). Sedentary activity (min/d) was calculated with the formula: 1440 – Sleep – LPA − MVPA intensity physical activity, where the value 1440 represents the total number of daily minutes. Average daily METs were calculated by summing the MET values for each 1-min epoch and then dividing it by the number of minutes of valid data for each day.

Resting energy expenditure (REE, kcal/d) was estimated based on the SWAM data acquired in night recording of sleep. The energy expenditure during 1 h of early sleep was extrapolated to 24 h according to the manufacturer’s instructions. Net activity energy expenditure (AEE, kcal/d) was computed as activity energy expenditure calculated by the SWAM minus the subject’s and relative REE. TEE (kcal/d) was calculated as the sum of REE and AEE, based on 10% of “diet-induced thermogenesis” [31]. Physical activity level (PAL)—an indicator of physical activity obtained from metabolic parameters—was calculated by dividing TEE by REE.

### 2.6. Exercise Training Protocols

On weekdays, the subjects trained three times a week for 8 weeks for a total of 24 training sessions. Training sessions were separated by at least 48 h. For both groups, each supervised session was preceded by a standardized warm-up involving light aerobic exercise (10 min pedaling at 50 W) and dynamic stretching (5 min). The entire training session lasted from 45 to 60 min.

The HIIT group (EXP) performed 7 × 2 min bouts of cycling exercise (915 E, Monark, Varberg, Sweden) at 85%–95% of individual $\overset{\cdot }{V}$O_2max_ interspersed by 2 min of recovery at 40% of $\overset{\cdot }{V}$O_2max_. The control group (CTRL) performed aerobic training on a stationary bike or treadmill (20–30 min at moderate-intensity, at 46%–64% of $\overset{\cdot }{V}$O_2max_, 5–6 on a 10-point scale). In both groups, the mechanical workloads of aerobic training related to the percentage of $\overset{\cdot }{V}$O_2max_ were calculated using the individual oxygen consumption/load ratio ($\overset{\cdot }{V}$O_2_/W) of the warm-up before the incremental test and created using the oxygen consumption values measured in the last minute of each load. Heart rate/load ratio (HR/W) was computed to adjust mechanical workloads every 2 weeks [41]. During each exercise session, HR data were monitored with chest belts (Polar Team2 Pro, Polar Electro Oy, Kempele, Finland)

The subjects wore the SWAM during two training sessions; the average data was used to estimate exercise session energy expenditure (kcal/exercise session). No injuries or health disorders occurred during the study period and the study protocol was completed as planned. All subjects attended all exercise sessions.

### 2.7. Statistics

The sample size for this study was calculated using software G* Power ver. 3.1.5.1 (HHU Düsseldorf, Düsseldorf, Germany). With 80% power and 5% significance level, the study requires a total sample size of 24 subjects. The Shapiro–Wilk test was applied to test for data normality. After normalization of data by logarithmic transformation, the ANOVA parametric test for repeated measures and Fisher’s post hoc test were applied to investigate main data variations using time (PRE vs. POST, T1 vs. T2 vs. T3, WD vs. WE during T1 or T2, TD vs. NTD during T2) as within factor and group (EXP vs. CTRL) as between factor. Statistical significance was set at *p* < 0.05. Power was reported (a power of 0.80 is generally considered acceptable, Cohen, 1988) [42]. All analyses were performed using StatView software version 5.0.1(SAS, North Carolina, NC, US). Data are expressed as mean ± standard deviation (SD).

## 3. Results

### 3.1. Anthropometric and Physiological Variables Before and After Training (PRE vs. POST)

In both groups significant differences after 8 weeks of training were recorded for body weight (EXP: PRE 77.8 ± 10.4 kg, POST 76.8 ± 9.9 kg; CTRL: PRE 78.3 ± 8.9 kg, POST 77.2 ± 8.8 kg; *p* = 0.0262; Power = 0.622) and BMI (EXP: BMI PRE 26.5 ± 2.8 kg/m^2^, POST 26.2 ± 2.7 kg/m^2^; CTRL: PRE 26.8 ± 2.9 kg/m^2^, POST: 26.4 ± 3, kg/m^2^; *p* = 0.0072; Power = 0.820). No significant changes were recorded for the other anthropometric variables or WB fat.

There was a significant increase in both absolute and relative $\overset{\cdot }{V}$O_2max_ for both groups ($\overset{\cdot }{V}$O_2max_ (mL/min) *p* < 0.0001; Power = 0.999; $\overset{\cdot }{V}$O_2max_ (ml/kg/min) *p* < 0.0001; Power = 1.000) (Table 2).

No significant time-by-group interactions were found.

### 3.2. Changes in Physical Activity and Energy Expenditure Between the Three Time Points (T1 vs. T2 vs. T3)

Sleep, sedentary activity, and light intensity physical activity remained constant for both groups during the 6 months of the study. There was a statistically significant difference in moderate physical activity (*p* = 0.0115; Power = 0.788) and in vigorous physical activity (*p* < 0.0001; Power = 1.000) between the time points (Figure 2), with a significant increase during the 8 weeks of training over baseline (T1 vs. T2) values for both groups (MPA: *p* = 0.0058; VPA: *p* < 0.0001) and a significant decrease after the 8 weeks of training (T2 vs. T3) for both groups (MPA: *p* = 0.0159; VPA: *p* < 0.0001). The majority of awake time was spent in sedentary and light intensity physical activity (<3 METs). No significant time-by-group interactions were found.

A significant increase in moderate-to-vigorous physical activity (*p* = 0.0028; Power = 0.910) was recorded for both groups only during the training period and the increase was not maintained at 2 months after the end of training (EXP: 110 ± 30 min/d; 134 ± 45 and 94 ± 23 min/d; CTRL: 86 ± 30 min/d; 141 ± 42 and 110 ± 47 min/d; T1, T2, and T3, respectively). No significant time-by-group interaction was found.

Additionally, a significant increase in average daily METs (*p* = 0.0275; Power = 0.673) was observed for both groups (EXP: 1.42 ± 0.12 METs; 1.52 ± 0.18 METs; 1.42 ± 0.11 METs; CTRL: 1.44 ± 0.16 METs; 1.59 ± 0.28 METs; 1.45 ± 0.24 METs; T1, T2, and T3, respectively). No significant time-by-group interaction was found.

There was no significant change in REE and TEE in both groups between T1, T2, and T3; there was, on the contrary, a significant change in AEE (*p* = 0.0033; Power = 0.901) between the time points (Figure 3). No significant time-by-group interactions were found.

The average estimated energy expenditure was 184 kcal/exercise session for the EXP and 155 kcal/exercise session for the CTRL group. The physical activity level in the EXP group was 1.39 ± 0.07, 1.45 ± 0.01, and 1.33 ± 0.05, and for the CTRL group was 1.34 ± 0.06, 1.47 ± 0.19, and 1.37 ± 0.10, at T1, T2, and T3, respectively, with a significant difference (*p* = 0.0004; Power = 0.981) between time points. No significant time-by-group interaction was found.

For both groups, comparison of data from T1, T2, and T3, to determine changes in energy expenditure during training and to understand whether it was maintained at 2 months after the end of training, show a significant increase in AEE and PAL during training (AEE, *p* = 0.0083; PAL, *p* < 0.0013) and a significant decrease after training (AEE, *p* = 0.0014; PAL, *p* < 0.0002) (Figure 3).

### 3.3. Compensatory Effects Analysis (T1 vs. T2)

We compared the data recorded on weekdays versus weekend-days between T1 and T2 and the data recorded on training and non-training days during T2 to determine whether a compensatory effect occurred during the training period.

For both groups, overall analysis of physical activity, sleeping habits, and energy expenditure during T2, compared to T1, showed no significant differences in weekend values, indicating that the subjects maintained their usual lifestyle. Only a significant increase in vigorous physical activity (*p* = 0.0459; Power = 0.515) was noted during the weekends.

During the weekdays, significant greater variations in moderate (*p* = 0.0147; Power = 0.721), vigorous (*p* < 0.0001; Power = 0.999), and moderate-to-vigorous physical activity (*p* < 0.0001; Power = 0.999), determining a significantly greater change in AEE and TEE (P = 0.0125; Power = 0.703; P = 0.0185; Power = 0.683, respectively), were found for the EXP group when compared with the CTRL group. METs significantly increase for the EXP group and significantly decrease for the CTRL group (*p* = 0.0422; Power = 0.532). All the other parameters (activity energy expenditure, physical activity level, sleep, sedentary activity, light intensity activity) remained unchanged (Table 3).

During training days (WD, TD), compared to non-training days (WD, NTD), moderate physical activity was increased for both groups (*p* = 0.0099; Power = 0.779), as was vigorous physical activity (*p* = 0.0005; Power = 0.982), moderate-to-vigorous physical activity (*p* = 0.0026; Power = 0.917), average daily METs (*p* = 0.0004; Power = 0.989), AEE (*p* < 0.0001; Power = 1.000), TEE (*p* < 0.0001; Power = 1.000), and PAL (*p* < 0.0001; Power = 1.000). The changes were attributed to the training session and were equivalent for both groups. Only during T2 and analyzing the differences only during WD (TD and NTD) a significant difference in sleeping time was detected between EXP and CTRL groups (*p* = 0.0003; Power = 0.991), ascribable to a different sleeping habit at that time (Table 3).

A significant time-by-day-by-group interaction was found for VPA and METs, underlying the greater increase in VPA, MVPA and METs in T2 for the EXP group during WD (VPA: *p* < 0.0001, Power = 1.000; MVPA: *p* = 0.0173, Power = 0.694; METs: *p* = 0.0178, Power = 0.690, respectively).

### 3.4. Between-Group Differences

For each training session, the mean total duration of aerobic training was similar for the two groups (EXP: 26 ± 2 min; CTRL: 27 ± 3 min).

No statistically significant differences were observed between the two groups for the anthropometric and physiological variables before and after training (PRE vs. POST) or for the components of daily physical activity and energy expenditure before, during, and after training (at T1, T2, and T3, respectively). Improvements in cardiorespiratory fitness were observed for both groups and were attributable to the performed training.

Significant differences between the two groups were detected for levels of physical activity (VPA: *p* = 0.0182; MVPA: *p* = 0.0182) maintained during weekdays, with higher physical activity levels noted for the EXP group. A significant time-by-group interaction was observed for AEE, TEE, PAL, and VPA, indicating a larger increase in energy expenditure in the EXP group if compared to the CTRL group, due to the decrease of weekends values in T2 for the CTRL group (AEE: *p* = 0.0344, Power = 0.571; TEE: *p* = 0.0191, Power = 0.678; PAL: *p* = 0.0476, Power = 0.508; VPA: *p* = 0.0360, Power = 0.563, respectively).

## 4. Discussion

With the present study, we wanted to examine the effect of 8 weeks of HIIT on daily physical activity and energy expenditure compared to training at moderate intensity in healthy elderly men.

Our study is the first to investigate the compensatory effects of HIIT on physical activity and energy expenditure in older adults. Other strengths include (1) objective measurement of daily physical activity and energy expenditure using a multisensor activity monitor: our study highlights the ease of use of a SenseWear Armband Mini for evaluating physical lifestyle in older adults; (2) assessment of physical activity over a 6-day period; (3) analysis of the differences in physical activity patterns between weekdays and weekends, which need to be considered in view of changes in activity behavior between weekend days and weekdays [43]; (4) analysis of four levels of intensity of physical activity (sedentary, light, moderate, vigorous) that must be quantified in any research into the efficacy of interventions promoting healthy aging [44].

Our findings show a significant effect of training, with short sessions of exercise performed at high intensity, on moderate and vigorous physical activity, average daily METs, physical activity level, and activity energy expenditure, compared to baseline. At 2 months after the completion of HIIT, activity energy expenditure and physical activity levels returned at a similar level of baseline. A lack of difference in physical activity and energy expenditure between weekends, an increase in physical activity on training days, and no reduction on non-training days were observed for the experimental group. Furthermore, the increase observed between weekdays due to HIIT may argue against a compensatory effect for the experimental group. No change in physical activity and total energy expenditure during the weekend-days was noted also for the control group. However, physical activity, total energy expenditure, and METs were decreased on weekdays. Our data provide no evidence of a compensatory effect of HIIT, whereas seem to suggest a compensatory effect for subjects who performed longer, continuous, and moderate aerobic training.

Our data agree with some previous studies, but are not consistent with others. Studies examining the extent to which exercise interventions can modify total energy expenditure have reported that some people reduce their level of physical activity when they participate in an exercise program. Goran and Poehlman showed that in elderly adults training did not result in an increase in average energy expenditure: training was compensated by a corresponding reduction in non-training physical activity [23]. For instance, elderly participants engaged in vigorous training intensity (60%–85% $\overset{\cdot }{V}$O_2max_) felt fatigued and needed to rest for the remainder of the day. Wasenius and colleagues found a compensatory effect on lower intensity physical activity after structured aerobic exercise in middle-aged adults [24,25]. Meijer and colleagues observed the same compensatory effect of endurance-strength training in older adults who may anticipate the training program by lowering their non-training level of physical activity [26]. The study also suggested a compensatory effect induced by an additional amount of physical exercise but followed by decrease in energy expenditure and physical activity [27]. Conversely, Washburn and Ficker [29] and Hunter and colleagues [30] found no compensatory effect in older adults after structured vigorous aerobic exercise (≥6 metabolic equivalents, METs) or after resistance training, respectively. De Moura and colleagues found no compensatory effect in adults after aerobic exercise [28].

Our data showed a difference (*p* = 0.06) in total energy expenditure during training; this may be partially explained by a slight, albeit insignificant, increase in resting energy expenditure, as reported by Hunter and colleagues [30]. The significant increase in activity energy expenditure, caused by the increase in metabolic rate due to HIIT (≈184 kcal/session), was the main cause for the increase in total energy expenditure and physical activity level. Moreover, our data show that HIIT significantly affected vigorous physical activity on training days but not the general physical activity patterns on non-training days. This difference shows that participation in a vigorous exercise program does not alter the level of physical activity during non-exercise time in elderly adults [29]. Overall analysis of physical activity during HIIT showed that the subjects were able to maintain their routine lifestyle. This observation is shared by Washburn and Ficker and Hunter and colleagues who noted that older adults did not attenuate free-living physical activity during a vigorous training program [29,30].

While our data show a significant increase in levels of physical activity and daily energy expenditure during HIIT, these lifestyle changes were not maintained at 2 months after the end of the HIIT program. Our results contrast with a recent review that demonstrated how physical exercise interventions in adults aged 55–70 years may lead to long-term improvements in physical activity for up to 12 months [45]. This discrepancy is likely due to the fact that our subjects were fairly active before, during, and after the training period (moderate physical activity > 90 min/d) and maintained their physical activity levels three-fold higher than that recommended for older adults [46].

Since our results for the physiological and anthropometrical parameters obtained with HIIT have been reported and discussed elsewhere, it is sufficient here to say that they substantiate previous data showing that HIIT can improve cardiovascular performance with substantial positive effects on health-related parameters [8,10,11,47,48,49].

No drop outs were recorded during the training period (all subjects concluded the 24 training sessions and no training-related injuries occurred) for HIIT in particular, considering that the men appreciated the training and tolerated the high intensity exercise program. It can be stated that, compared to moderate training, HIIT is a good time-efficient workout strategy that stimulates adaptations in healthy older adults [50]. Nevertheless, our study has some limitations: (1) the small sample size of the two groups; (2) the 6 months’ duration of the study, which may imply seasonal variations in physical activity and energy expenditure; (3) diet was not monitored during the study.

## 5. Conclusions

The results of this study suggest that no compensatory effect occurred as a consequence of HIIT. Improvements in physical activity were observed for the experimental group and were attributable to the specificity of training, HIIT, with no significant changes in lifestyle habits noted. Our findings provide new evidence that HIIT does not adversely affect the lifestyle of active older adults, since it neither reduces daily energy expenditure nor increases sedentary time. In contrast, the control group modified lifestyle habits, increasing physical activity during training days, but decreasing the overall physical activity during weekdays, suggesting a need to refrain the day after practice.

Fundamental questions remain to be answered regarding the minimum volume of exercise necessary to improve physiological well-being, the effectiveness of alternative interval-training strategies, and the precise nature and magnitude of adaptations in lifestyle that can be elicited and maintained long-term.

## Figures and Tables

**Figure 1 ijerph-17-01083-f001:**
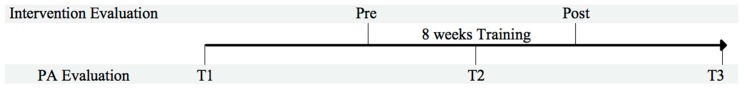
Assessment timeline; PA (Physical activity) evaluation; T1: Evaluation before the training; T2: Evaluation during training T3: Evaluation after training.

**Figure 2 ijerph-17-01083-f002:**
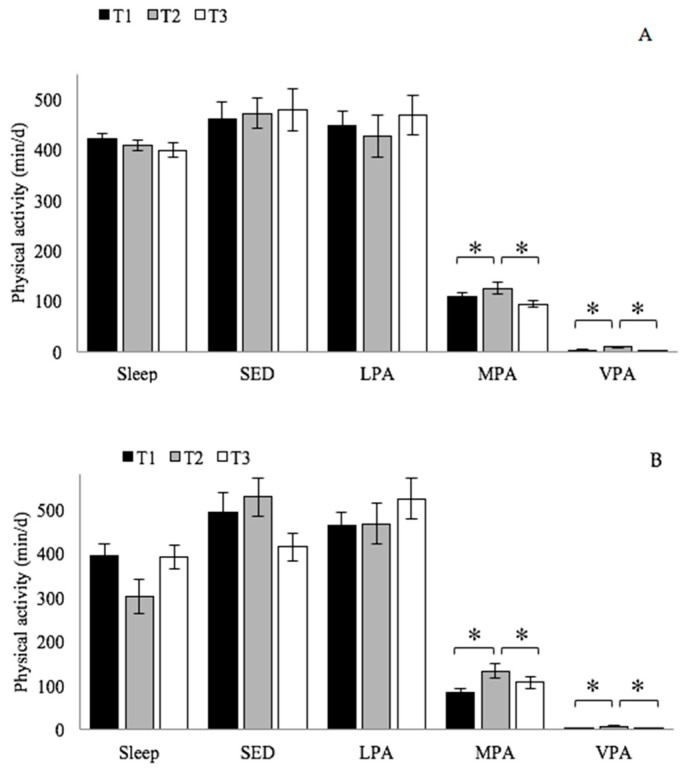
Mean of components of daily physical activity (Sleep, sleep time; SED, sedentary activity, intensity < 1.5 METs; LPA, light intensity physical activity, 1.5–3 METs; MPA, moderate intensity physical activity, 3–6 METs; VPA intensity physical activity > 6 METs) before the training period (T1), one week during training period (T2), and one week after two months of the end of the training (T3). Repeated measure Anova (* Significantly different, *p* < 0.05). (**A**) EXP group; (**B**) CTRL group.

**Figure 3 ijerph-17-01083-f003:**
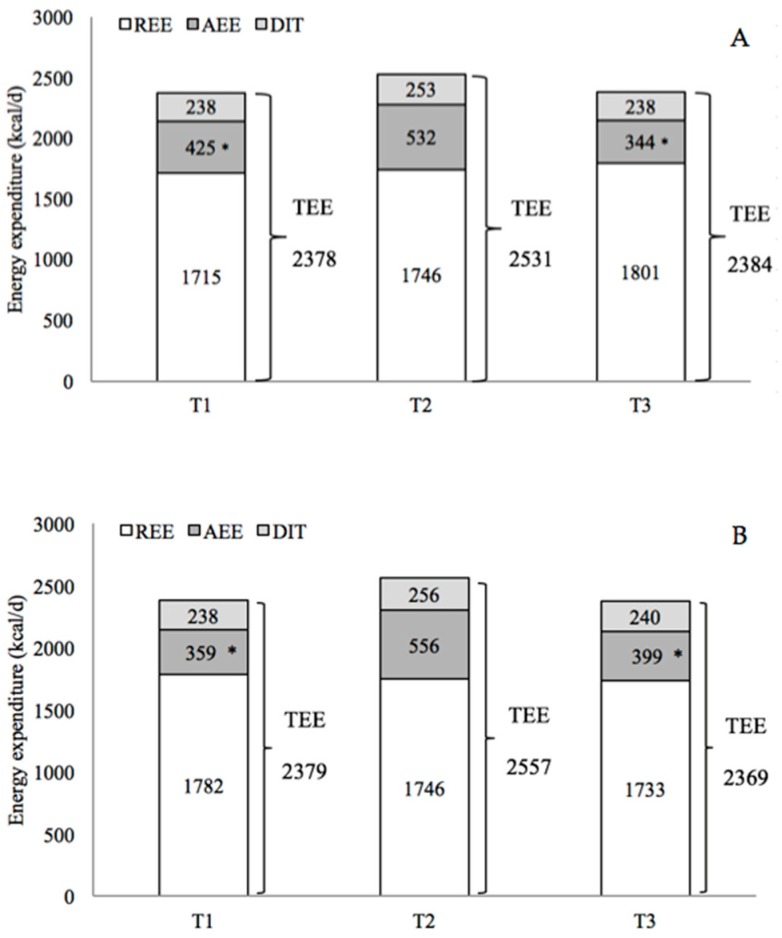
Mean of components of total energy expenditure (TEE) before the training period (T1), one week during training (T2), and one week after two months of the end of the training (T3). TEE: total energy expenditure; REE: resting energy expenditure; AEE: activity energy expenditure; DIT: Diet induced thermogenesis. Repeated measure Anova (*Significantly different compared to T2, *p* < 0.05). (**A**) EXP group; (**B**) CTRL group.

**Table 1 ijerph-17-01083-t001:** Participant characteristics (mean ± SD).

	EXP	CTRL
Age (years)	69.4 ± 4.3	69.67 ± 4.1
Height (cm)	171.2 ± 5.3	171 ± 7.2
Weight (kg)	77.8 ± 10.4	78.3 ± 8.9
BMI (kg/m^2^)	26.5 ± 2.8	26.8 ± 2.9

BMI: body mass index; EXP: experimental group; CTRL: control group.

**Table 2 ijerph-17-01083-t002:** Anthropometric and physiologic variables pre- and post-intervention (mean ± SD).

	PRE EXP	POST EXP	VAR %	PRE CTRL	POST CTRL	Var %
WC (cm)	95.2 ± 9.7	93.9 ± 9.2	−1.4	98.2 ± 8.5	98.3 ± 8.1	0.1
WHR	0.92 ± 0.07	0.92 ± 0.06	0.0	0.95 ± 0.05	0.95 ± 0.05	0.0
WHtR	0.56 ± 0.05	0.55 ± 0.05	−1.8	0.58 ± 0.06	0.58 ± 0.05	0.0
WB fat (%)	26.4 ± 5.9	25.2 ± 6.2	−4.5	25.5 ± 4.3	25.6 ± 5.3	0.4
$\overset{\cdot }{V}$O_2max_ (mL/min)	2310 ± 343	2485 ± 384 *	7.6	2186 ± 326	2352 ± 338 *	7.6
$\overset{\cdot }{V}$O_2max_ (mL/kg/min)	29.9 ± 4.3	32.7 ± 6.0 *	9.2	28.3 ± 6	30.8 ± 6.1 *	8.8

* significant difference compared to pre-training. (Repeated measure Anova, *p* < 0.05). Var %: Percentage difference Post–Pre; WC: waist circumference; WHR: waist to hip ratio; WHtR: waist to height ratio; WB fat: whole body fat; $\overset{\cdot }{V}$O_2max_: maximal oxygen uptake.

**Table 3 ijerph-17-01083-t003:** Energy expenditure and physical activity maintained during weekly (WD) and weekend (WE) days in T1 and T2. In T2, weekdays were split into training (WD, TD) and non-training (WD, NTD) days (mean ± SD).

		T1		T2			
		WE	WD	WE	WD	WD, TD	WD, NTD
EXP	SED (min/d)	423 ± 116	472 ± 140	410 ± 110	458 ± 109	561 ± 140	500 ± 122
	LPA (min/d)	488 ± 115	434 ± 132	513 ± 150	429 ± 144	426 ± 148	522 ± 147
	MPA (min/d)	114 ± 46	105 ± 31	100 ± 48	137 ± 53 *	161 ± 66 ^$^	114 ± 55
	VPA (min/d)	2 ± 2	3 ± 2	4 ± 6 ^#^	11 ± 6 *^, †^	15 ± 12 ^$^	6 ± 6
	MVPA (min/d)	116 ± 47	108 ± 32	104 ± 51	149 ± 55 *^,^^†^	176 ± 70 ^$^	120 ± 59
	Sleep (min/d)	431 ± 44	427 ± 43	413 ± 71	405 ± 44	277 ± 134	298 ± 115
	Av. daily METs (MET)	1.44 ± 0.16	1.41 ± 0.12	1.42 ± 0.17	1.57 ± 0.21*	1.7 ± 0.28 ^$^	1.55 ± 0.25
	AEE (kcal/d)	421 ± 148	427 ± 130	383 ± 149	606 ± 210 *	741 ± 299 ^$^	467 ± 221
	TEE (kcal/d)	2373 ± 229	2379 ± 285	2365 ± 240	2613 ± 332 *	2763 ± 442 ^$^	2457 ± 341
	PAL	1.38 ± 0.11	1.38 ± 0.07	1.36 ± 0.10	1.50 ± 0.13 ^†^	1.58 ± 0.17 ^$^	1.41 ± 0.14
CTRL	SED (min/d)	459 ± 138	457 ± 112	483 ± 119	468 ± 133	455 ± 105	465 ± 106
	LPA (min/d)	459 ± 145	439 ± 138	509 ± 165	461 ± 161	432 ± 145	430 ± 93
	MPA (min/d)	102 ± 40	115 ± 41	119 ± 56	92 ± 45	133 ± 61 ^$^	107 ± 25
	VPA (min/d)	2 ± 1	5 ± 5	7 ± 8 ^#^	4 ± 2	9 ± 6 ^$^	3 ± 3
	MVPA (min/d)	104 ± 40,	120 ± 45	127 ± 62	94 ± 47	142 ± 64 ^$^	111 ± 27
	Sleep (min/d)	418 ± 46	425 ± 54	321 ± 128	418 ± 85	411 ± 53	434 ± 41
	Av. daily METs (MET)	1.43 ± 0.13	1.51 ± 0.16	1.52 ± 0.22	1.42 ± 0.17 *	1.58 ± 0.22 ^$^	1.41 ± 0.15
	AEE (kcal/d)	522 ± 199	441 ± 145	391 ± 186	458 ± 175	602 ± 198 ^$^	445 ± 104
	TEE (kcal/d)	2513 ± 228	2453 ± 159	2361 ± 197	2382 ± 233	2594 ± 315 ^$^	2371 ± 270
	PAL	1.46 ± 0.16	1.40 ± 0.12	1.37 ± 0.13	1.42 ± 0.14	1.50 ± 0.12 ^$^	1.41 ± 0.07

Repeated measure Anova: ^#^ Significant difference in weekend days (WE) between T1 and T2. * Significant difference in weekdays (WD) between T1 and T2. ^$^ Significant difference in weekdays with or without training (WD, TD vs. WD, NTD) in T2. ^†^ Significant difference between groups (EXP vs. CTRL) in WD, T2. SED: time spent in sedentary activity; LPA: light physical activity; MPA: moderate physical activity; VPA: vigorous physical activity; MVPA: moderate and vigorous physical activity; Sleep: time spent sleeping; AEE: activity energy expenditure, TEE: total energy expenditure, PAL: physical activity level.

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
