# Peer review of "High Intensity Interval Training Does Not Have Compensatory Effects on Physical Activity Levels in Older Adults"

_ijerph, 2020, doi:10.3390/ijerph17031083_

Round 1
Reviewer 1 Report
Line 40: Sounds awkward: And since…
Line 44: reference location need to be at the end of this sentence not after author’s name
Line 51-69: Please summarize this paragraph to make it short since it seems more appropriate to be in discussion section.
Line 82, 83: can IRB approval number be included here?
Line 87: during the post – T3, what were the participants supposed to do? Were they asked to maintain their improved level of physical activity? Or asked to stop?
Line 94: Not sure if ‘staggered’ is a proper term to use
Line 142: What is the evidence to categorize this four intensity level? Please reference the resources
Line 163: How is individual % VO2max identified during the HIIT session? Also how much was the total duration of training session different between groups?
Line: 176: Please show how 24 patients came up based on the sample size calculation. Also this type of research question need to be analyzed by repeated measures ANOVA and paired t-test.
Table 2: please report P-value for within group difference and between group difference
Line 220-223: this paragraph also need to be done by repeated measures ANOVA to show group x time interaction.
Table3: where are the data for T3?
Line 276: unclear why no data on T3 reported here?
Discussion and conclusion need to be revised significantly because the author’s research question has to be answered by the statistics of group x time interaction (repeated measures ANOVA), not just within group or between group difference separately.
Reviewer 2 Report
Overall this is an interesting study but there are a number of areas that need to be addressed.
Introduction
line 39 a reference is needed to support the statement that any physical activity no matter how small improves health
line 42-50 there have been a number of studies done in older adults using high intensity exercise for health, these should be explored
Methods
Major issue: the statistical analysis is unclear. Given that the data is not normally distributed then log transform the data to normalise it. This will allow for a more robust statistical analysis given the study design. I remain to be convinced that performing non parametric t-test for each variable is a suitable analysis. Another option would be to calculate percentage change across the time periods and then analyse this between groups. You cannot use Cohen's d for non parametric data.
Table 1 should be formatted consistently and the median values removed
line 97-100 A justification of using the physical activity monitor for 1 week at a time should be provided. Details of any controls used to ensure each time point was similar.
line 111 please provide the initial resistance for the incremental test
line 186-187 please use the definition for cohen's d that reflects training adaptations (Rhea, M.R. Determining the magnitude of treatment effects in strength training research through the use of the effect Size. Strength Cond. Res. 2004, 18, 918–920)
Results
Please remove the median values throughout the results section and only report group mean data. The median value is not telling us anything worth knowing with respect to the adaptations being looked at. Also put a zero in front of a decimal point when reporting p values or effect sizes.
Figure 2: please add error bars and consider breaking the graph up as you cannot see VPA.
line 216 please remove the term non significant increase.
Discussion
This section may need to change depending on reanalysis of the data.
line 304-314 please remove this paragraph, it is not a discussion of the findings. If you wish to show validity of technology then this should be done in the methods section.
Round 2
Reviewer 1 Report
Thank you for the revised version.
Reviewer 2 Report
The authors have addressed all concerns and the paper has been improved.